# Impact Toughness Anisotropy of TA31 Titanium Alloy Cylindrical Shell after Ring Rolling

**DOI:** 10.3390/ma13194332

**Published:** 2020-09-29

**Authors:** Haiyang Jiang, Jianyang Zhang, Bijun Xie, Zhangxun He, Hao Zhang, Bing Wang, Bin Xu, Yuxi Wu, Mingyue Sun

**Affiliations:** 1Key Laboratory of Nuclear Materials and Safety Assessment, Institute of Metal Research, Chinese Academy of Sciences, Shenyang 110016, China; hyjiang19b@imr.ac.cn (H.J.); jyzhang15b@imr.ac.cn (J.Z.); bjxie@imr.ac.cn (B.X.); bwang17b@imr.ac.cn (B.W.); yxwu@imr.ac.cn (Y.W.); mysun@imr.ac.cn (M.S.); 2School of Materials Science and Engineering, University of Science and Technology of China, Hefei 230026, China; 3Shenyang National Laboratory for Materials Science, Institute of Metal Research, Chinese Academy of Sciences, Shenyang 110016, China; 4Wuhan Second Ship Design and Research Institute, Wuhan 430064, China; zhangxun_he@126.com (Z.H.); zhyf1408@126.com (H.Z.)

**Keywords:** TA31 titanium alloy cylindrical shell, ring rolling, impact toughness anisotropy, microstructure, α phase orientation

## Abstract

The impact toughness of a TA31 titanium alloy cylindrical shell was investigated systemically after ring rolling. The impact toughness of specimens with different notch orientations shows obvious anisotropy. The microstructure of the cylindrical shell and the impact fracture were characterized by an optical microscope and scanning electron microscope. The results show that cracks are easier to propagate in the equiaxed α phase than the elongated α phase. This is because the expanding cracking path in the equiaxed α phase is shorter than that in the elongated α phase, and thereby the cracks are easier to propagate in the equiaxed α phase than the elongated α phase. More specifically, the α phase on the RD-TD plane was obviously isotropic, which makes it easy for the cracks to propagate along α grains in the same direction. However, the α phase on the RD-ND plane has a layered characteristic, and the direction of the α phase varies from layer to layer, thus it requires higher energy for cracks to propagate across this layered α phase. Therefore, the cracks propagating in the same α phase orientation take easier than that in the layered α phase, so it has lower impact toughness.

## 1. Introduction

Titanium alloy is widely used in manufacturing key components for aerospace engineering, due to its high specific strength, low density, superior corrosion resistance and good process performance [1,2,3,4,5]. Titanium alloy with a composite microstructure of equiaxed α and transformed β, in which the α phase accounts for 50%, generally has excellent ductility and fatigue resistance [6,7,8,9]. Nakase et al. [10] studied the influence of the microstructure parameters produced by α+β forging and heat treatment on the strength and ductility of Ti-6Al-4V. They suggested that the strength and ductility have strong correlations with the aspect ratio of the primary α phase and increase linearly with the decrease in the aspect ratio of the primary α phase. In addition, Qin et al. [11] found that the full lamellar microstructure of Ti-5553s alloy can transform into a bimodal microstructure during α+β forging. They provided a theoretical basis for the forging of the alloy and clarified the relationship between the microstructure and mechanical properties. Other processing methods can also effectively change the structure and properties. Kulagin et al. [12] discovered severe plastic deformation (SPD) processing can open up new avenues for creating advanced titanium alloy with outstanding mechanical properties. Estrin et al. [13] utilized SPD technology to obtain the ultrafine-grained entangled spiral rods of Ti-6Al-4V. 

Since the α phase of the hexagonal close-packed structure has obvious anisotropy, the mechanical properties of the strong texture structure change significantly with the change of the loading direction [14,15,16]. Zhao et al. [17] investigated the evolution of the microstructure and texture of Ti65 alloy sheets developed by unidirectional rolling (UDR) and cross rolling (CR) followed by solution and aging treatment. The result provided a reference for precise control of the microstructure of Ti65 alloy sheet. Bache et al. [14] performed a systematic research on the effect of texture on the mechanical properties of Ti-6Al-4V alloy sheets, and indicated that the yield and tensile strength of the transverse specimens of the sheet were higher than those of the rolling direction specimens, and the fatigue life of the transverse specimens was lower than the specimen of rolling direction. Zhu et al. [18] studied the relationship between the texture type and mechanical properties of cold-rolled and annealed industrial pure titanium sheets. Their results proved that the {0001} base texture can improve the plasticity of the sheet, and the transverse tensile strength of the sheet was higher. Wang et al. [19] found that rotary swaging (RS) and the following annealing treatment have great effect on the grain refinement and texture, and elucidated the mechanism of the ultrafine-grained microstructure and strong fiber texture of improving mechanical properties. Impact toughness is an important index to evaluate the ability of materials to absorb plastic deformation under impact load, and to reflect the materials’ impact resistance [20]. Impact toughness is closely related to the orientation of the α phase in titanium alloys. Huang et al. [21] investigated the influence of anisotropy on the microstructure and mechanical properties of Ti/Al laminated composites fabricated by rolling, and found that the microstructure and mechanical properties of the Ti/Al laminated composites were obviously anisotropic due to grain orientation and twins. Zarkades et al. [22] investigated titanium plate including commercially pure Ti-6Al-4V, Ti-4Al-4V, Ti-4Al-4Mn and Ti-8Mn. They revealed that significant toughness variations can be found in titanium plate as a function of specimen and notch orientation. Hrabe et al. [23] found that Charpy absorption energy is affected by different heat treatment conditions, crystal orientations and anisotropic crystal grain morphologies. Grell et al. [24] reported the influence of powder oxidation and specimen orientation on the Charpy impact energy of Ti-6Al-4V parts manufactured using electron beam melting and found that the influence of specimen orientation is more serious than that of oxidation. The orientation of the α phase is not only the parameter to effect mechanical properties, but the morphology of the α phase also has an important effect on mechanical properties. Zhu et al. [25] analyzed the microstructure differences between hot-rolled and heat-treated titanium alloys and their influences on the ballistic impact behavior of the hot-rolled and heat-treated titanium alloys, and revealed the failure mechanism of the hot-rolled and heat-treated titanium alloy plates. Buirette et al. [26] studied the effect of the microstructure and macroscopic texture on Ti-6Al-4V Charpy impact toughness. However, the mechanism of the effects on impact toughness is not clear, and the related research is scare. 

Therefore, in this study, a near-α titanium alloy TA31 cylindrical shell manufactured by forging and ring rolling was selected. Its impact toughness was also tested. The microstructure of the cylindrical shell and the α phase orientation near the notch of the impact fracture were characterized in detail to reveal the effect of the microstructure and texture on the impact toughness. 

## 2. Materials and Procedure

### 2.1. As-Received TA31 Cylindrical Shell 

In this study, commercial-grade TA31 titanium alloy (Western Superconducting Technologies Co., Ltd., Xi’an, China) with the following chemical composition (wt%): Nb 2.87, Mo 1.00, Zr 2.05, Al 6.03, Si 0.021, Fe 0.025, C 0.0008 and Ti bal., was selected as the starting material. The cylindrical shell used in this study was manufactured from a forging rod with a diameter of 500 mm through upsetting, punching, saddle forging, core bar stretching and ring rolling (Figure 1a–e). The final size of this cylindrical shell was 900 mm in diameter, 500 mm in height and 40 mm in thickness (Figure 1f). The temperature range of each hot deformation process was strictly controlled at 960–800 °C. Finally, the entire forging was annealed at 900 °C for 2 h followed by air cooling to room temperature (RT). 

### 2.2. Impact Toughness Tests

The impact specimens of two different notch orientations were taken from the 1/2 thickness of the cylindrical shell, as shown in Figure 2a. In order to ensure the reproducibility of the results, at least three specimens were tested for each condition. The location of sampling and the notch orientations of the impact toughness specimens are shown in Figure 2b. Specimens of the TA31 alloy impact toughness tests were taken along the RD direction with two different notch orientations (RD-TD and RD-ND). The first two letters in the RD-TD name refer to the normal direction of the fracture, and the last two letters refer to the direction parallel to the crack propagation direction. 

The geometry of the impact specimen in this experiment is shown in Figure 2c. A standard specimen with a “V-shaped” notch is used. These “V-shaped” notch specimens can locate the fracture plane precisely, limit the propagation of the shear lip, and force the propagation of the crack along the width of the notch under plane strain conditions. Impact toughness tests were carried out on a ZBC2452-C pendulum impact testing machine with a load capacity of 450 J and test speed of 5.24 m/s at RT.

### 2.3. Microstructure and Orientation Characterization

The impact specimens were cut from the center along the lengthwise direction for the initial microstructure to be examined (Figure 3). The specimens were ground and polished according to standard procedures. The polished specimens were chemically etched in a hydrofluoric acid solution (47 mL H_2_O + 2 mL HNO_3_ + 1 mL HF) (Binzhou Guangyou Chemical Co., Ltd., Binzhou, China). Then, the microstructures were investigated with an optical microscope (OM) (Zeiss, Axio Observer A1m, Jana, German). The fracture morphology was characterized by scanning electron microscope (SEM) Electron back-scattered diffraction (EBSD) measurements were carried out with an HKL Nordlysnano detector (Oxford Instruments, Abingdon, Oxfordshire, UK ) equipped on a ZEISS MERLIN Compact instrument at step sizes of 1.5 µm depending on the grain size being analyzed. Automated orientation analyses were performed with the Channel 5 software package (Oxford Instruments, Abingdon, Oxfordshire, UK).

## 3. Results

### 3.1. The Microstructure of TA31 Cylindrical Shell

The α phase exhibits a bright contrast under the optical microscope, while the contrast of the β phase is dark (Figure 4). In general, the majority of the α phase is primary equiaxed α phase, while the transformed β matrix is composed of the β phase and secondary α lamella. The microstructure of the RD-TD plane is composed of 50% equiaxed α phase, and its size is about 20 μm. The rest is made of small strips of α platelets (1 μm thick) that are separated by thin β layers (Figure 4a). This is a typical microstructure of titanium alloy formed using α+β forging. As the main deformation direction is normal to the RD-TD plane, the microstructure on this plane does not show obvious deformation. However, on the RD-ND plane, insufficient deformation causes the α phase to be elongated along the tangential direction. Only part of the α phase is equiaxed, and its microstructure shows obvious deformation (Figure 4b). In the process of hot deformation of titanium alloys, the recovery and recrystallization of the α phase are important processes to keep α equiaxed. The spheroidization of the α phase is the most important microstructure evolution phenomenon for forging in the α + β phase [27,28,29]. 

### 3.2. Anisotropy of the Impact Toughness

The impact toughness test results are shown in Figure 5. Three specimens were tested for each notch orientation. The average impact energy of the RD-ND specimen is 91 J, which is higher than that of the RD-TD specimen (51 J). The impact energy of the three parallel specimens in each group is not much different. A strong anisotropy of the impact energy is shown for different notch orientations. 

### 3.3. Fractography

From the metallographic microstructure of the impact fracture near the notch (Figure 6), specimens selected from the RD-TD and RD-ND specimens have different α phase morphologies (Figure 6a,b). The main propagation mode of the cracks is intergranular fracture, accompanied by transgranular fracture. For the RD-TD specimen, the cracks mainly travel along the equiaxed α grain boundary, and the energy required is less than the propagation mode of a transgranular fracture. Hence, the impact energy is small with the flat crack propagation path and the impact toughness is low. Correspondingly, for the RD-ND specimen, when the crack passes through the elongated α phase, a large amount of energy is absorbed. Thus, the impact energy is higher with the bending crack propagation path and the impact toughness is also higher [30].

Impact fracture surfaces of the RD-TD and RD-ND specimens were observed. The crack initiation and propagation area were obtained by observing the macroscopic surface of the fracture. It can be seen that the crack originated from the notch and further expanded until it ruptured. Figure 7a–c show the morphology of the macroscopic fracture in the RD-TD specimen. Those fracture surfaces are relatively flat. In contrast, the surface of the macroscopic fracture in the RD-ND specimen shows large undulations, and tear edges perpendicular to the crack propagation direction were found (Figure 7d–f). 

The morphology of the fracture (Figure 8) shows that the dimples have a certain directionality (yellow dotted line). This related to the morphology of the α phase. Figure 8a shows the morphology of the fracture on the RD-TD specimen. The microfracture is composed of equiaxed dimples. Figure 8b is the morphology of the fracture on the RD-ND specimen. A lot of dimples are large and uniform. The undulating steps on the fracture are formed because the cracks cannot pass straight through the grains during the process of crack propagation. 

## 4. Discussion

### 4.1. Influence of the Microstructure on the Impact Energy

By observing the microstructure and fracture morphology of the cylindrical shell in different directions, it is obvious that the microstructure has a strong influence on the impact energy value. This difference in impact energy values is mainly attributed to the different crack propagation paths caused by the grain shape [31]. Figure 9 shows the overall morphology of the fracture surface after multiple images of the fragment. The specimen with an elongated α phase has a longer crack propagation path compared to the specimen with an equiaxed α phase, while the fracture surface of the specimen with an equiaxed α phase microstructure is relatively flat, and the crack propagation path is also shorter. Obviously, the microstructure with an elongated α phase has larger undulations on the fracture surface and has a longer crack propagation path. 

The crack propagation process in specimens with different grain shapes is clearly illustrated in Figure 10. The crack propagation path is shorter in the equiaxed α phase specimen. This is mainly attributable to the relatively weak obstructive effect of equiaxed α on cracks [32]. Cracks tend to propagate easily in the equiaxed α phase, as shown in Figure 10a. Therefore, the impact fracture of this specimen has the characteristics of short crack path length and low impact toughness. In contrast, the path of crack propagation is longer and rougher in the elongated α phase specimen compared to the equiaxed specimen, as shown in Figure 10b. The cracks of this specimen are difficult to propagate during testing, and thus have higher impact toughness.

### 4.2. Effect of the Orientation on Crack Propagation Behavior during Impact Toughness Tests

EBSD analysis was performed on the cross-sections of impact specimens with RD-TD and RD-ND. The test area was near the beginning of the impact crack, and the red box area was the test position. The test results of the RD-TD specimen are shown in Figure 11. Most of the α grain orientations in the test area are in the same direction. These areas correspond to the equiaxed α phase on the RD-TD plane. The α grain orientation can be clearly seen in the enlarged EBSD map in Figure 11b, and the hexagonal α phase exhibits a c-axis parallel to the ND direction.

As shown in Figure 12 (RD-ND specimen), the α grain distribution in this part occurs stratification. This indicated that the deformation in this direction during the forging process was relatively insufficient. A longer path was required for cracks to cross α grains of a different stratification [33]. Therefore, a large amount of energy can be absorbed when the crack grows. The impact value of the impact specimen sampled in the RD-ND direction is relatively high.

The crack propagation mode is determined by the distribution of the α grain orientation in the specimen. When the c-axes of the α grains are parallel to each other and perpendicular to the fracture direction, the cracks tend to propagate directionally between the layers of the α grains. The unstable voids between the layers will expand or crack with the plane strain during the plastic deformation process, resulting in rapid crack propagation. RD-ND specimens are generally regarded as composite materials with the same orientation and different orientations of the α phase (Figure 13) [4]. There are many layers in this specimen with the same orientation of the α phase. The cracks will propagate along the grain layer with the same orientation. When encountering the next grain layer with the same orientation, the propagation path of the crack will change, which leads to the need for higher energy for crack propagation [34]. 

The different fracture patterns are clearly illustrated (Figure 13). The α phase in the direction of the RD-TD notch forms regions in the same direction, and cracks can easily pass through these regions [34]. Therefore, it has lower impact toughness. In addition, in the RD-ND notch orientation specimens, the α phase has the characteristic of stratification. The crack will choose a more difficult path during the fracturing process. Cracks tend to propagate by bypassing different delamination α directions instead of directly passing through them. The cracks hardly deviate and produce a long crack path, so it has higher impact toughness.

## 5. Conclusions

The impact toughness of a TA31 titanium alloy cylindrical shell was investigated systemically after ring rolling. The impact toughness of specimens with different notch orientations shows obvious anisotropy. The following conclusions can be made:

(1) The TA31 titanium alloy cylindrical shell was formed by upsetting, punching, saddle forging, core bar stretching and ring rolling. The impact energy of the specimens with the notch facing the axial direction is generally higher than specimens with the notch facing the radial direction.

(2) The equiaxed α phase is presented in the RD-ND plane of the cylindrical shell, and the orientation of the α phase tends to be the same. However, the elongated α phase is formed in the RD-TD plane, and the orientation of the α phase occurs stratification.

(3) The impact energy of RD-ND specimens is higher than that of RD-TD specimens. On the one hand, cracks are easy to propagate in the equiaxed α-phase, which is related to very weak crack resistance, while the elongated α-phase has better crack propagation resistance. On the other hand, the α-phase in RD-TD specimens is obviously isotropic, causing cracks to easily propagate between α grains of the same orientation. The α phase in the RD-ND specimen has a layered characteristic, and the orientation of the α phase is different from layer to layer. When cracks pass through the α phase layers, higher energy is usually required.

## Figures and Tables

**Figure 1 materials-13-04332-f001:**
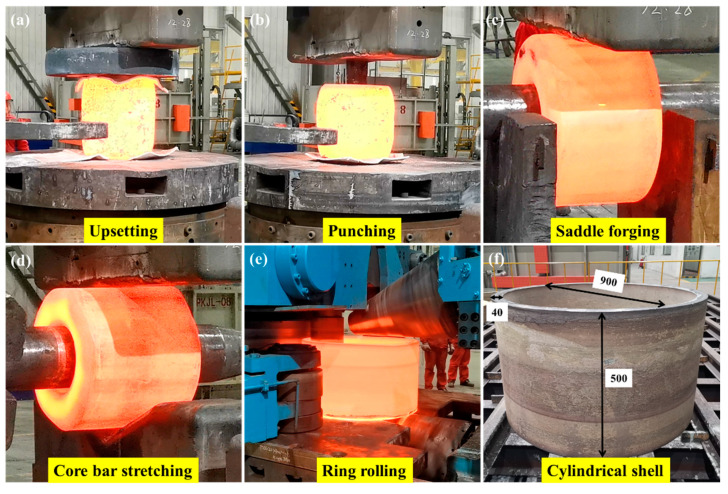
The forging process of the TA31 cylindrical shell through (**a**) upsetting, (**b**) punching, (**c**) saddle forging, (**d**) core bar stretching and (**e**) ring rolling; and (**f**) the size of the cylindrical shell (units is mm).

**Figure 2 materials-13-04332-f002:**
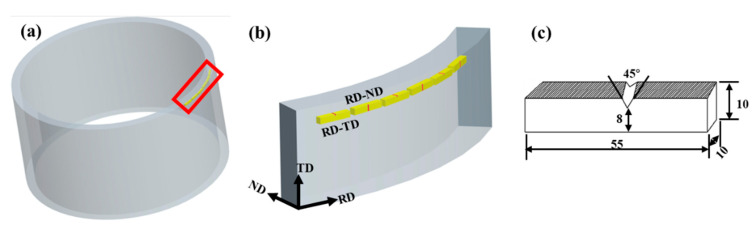
(**a**) The position of impact specimens in the cylindrical shell. (**b**) An enlarged view of the red box marked in (**a**). (**c**) The geometry of the impact specimen used in testing (units is mm). (RD represents the tangent direction, ND represents the radial direction and TD represents the axis direction).

**Figure 3 materials-13-04332-f003:**
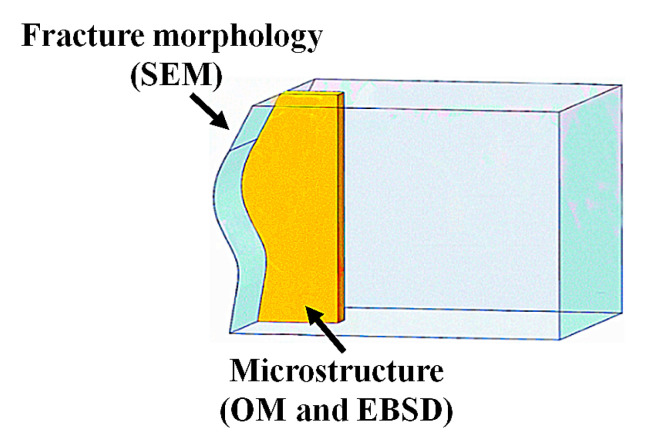
Schematic diagram of specimen locations in the OM, SEM and EBSD.

**Figure 4 materials-13-04332-f004:**
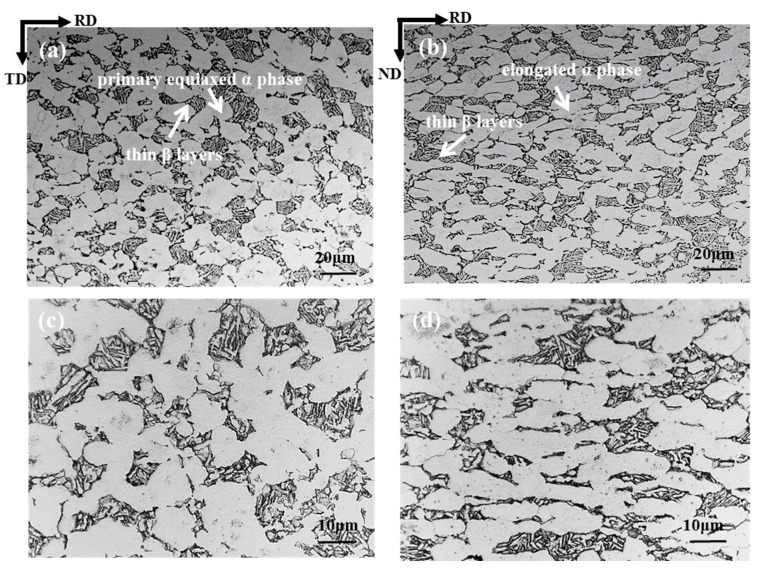
The initial microstructure of the TA31 cylindrical shell in different positions: (**a**,**c**) RD-TD plane; (**b**,**d**) RD-ND plane.

**Figure 5 materials-13-04332-f005:**
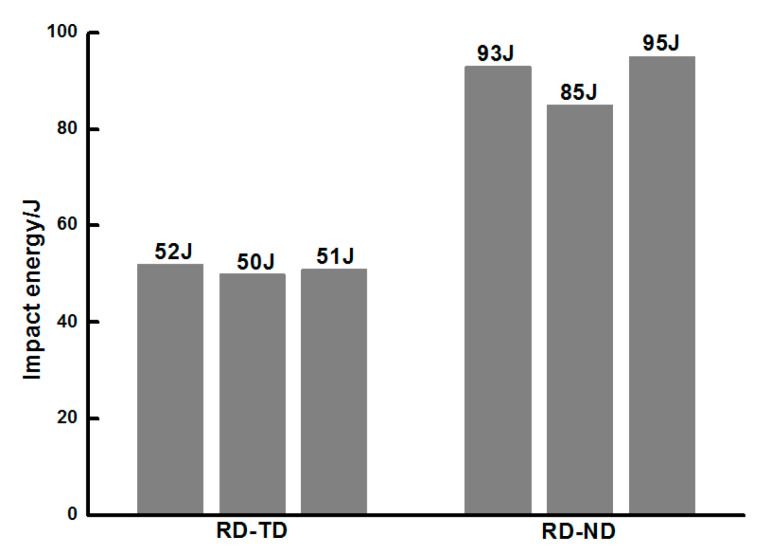
Impact energy for the RD-TD and RD-ND specimens.

**Figure 6 materials-13-04332-f006:**
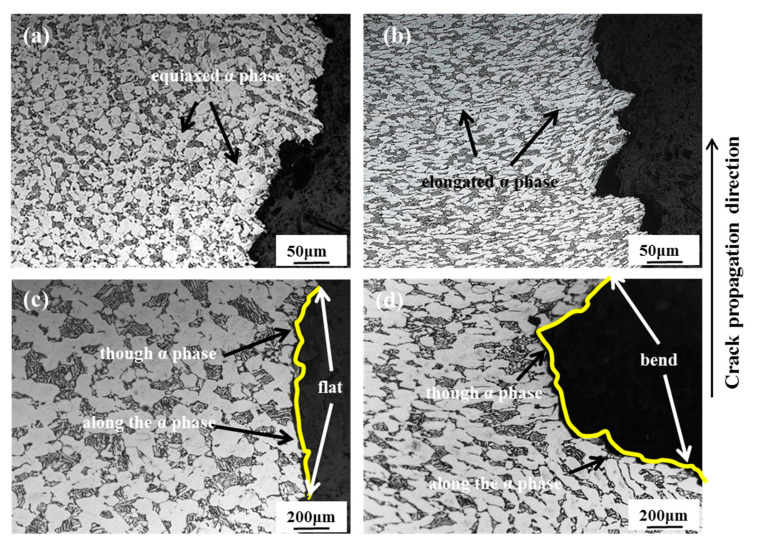
Metallographic microstructure of fractures for different notch orientations: (**a**,**c**) RD-TD specimen; (**b**,**d**) RD-ND specimen.

**Figure 7 materials-13-04332-f007:**
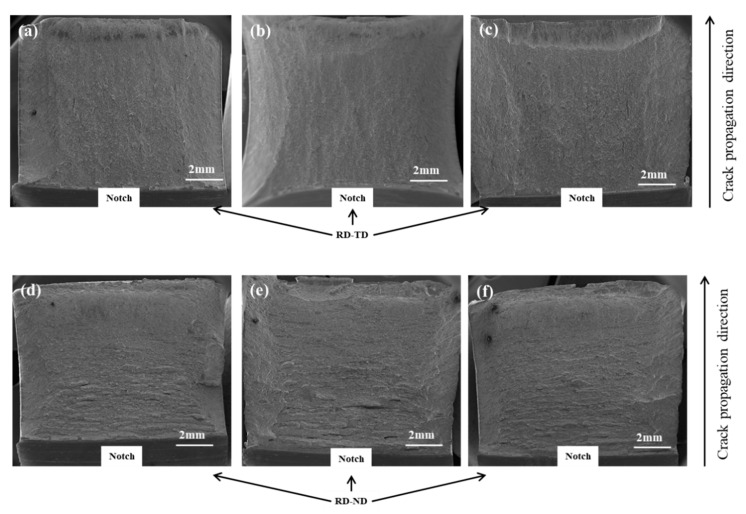
Macromorphology of the impact fractures: (**a**–**c**) RD-TD specimen; (**d**–**f**) RD-ND specimen.

**Figure 8 materials-13-04332-f008:**
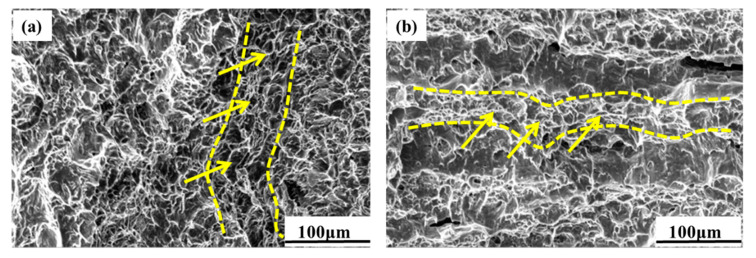
Microtopography of the impact fractures: (**a**) RD-TD specimen; (**b**) RD-ND specimen.

**Figure 9 materials-13-04332-f009:**
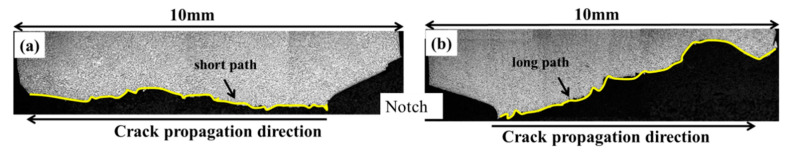
The representative crack propagation path in the impact specimens: (**a**) RD-TD specimen; (**b**) RD-ND specimen.

**Figure 10 materials-13-04332-f010:**
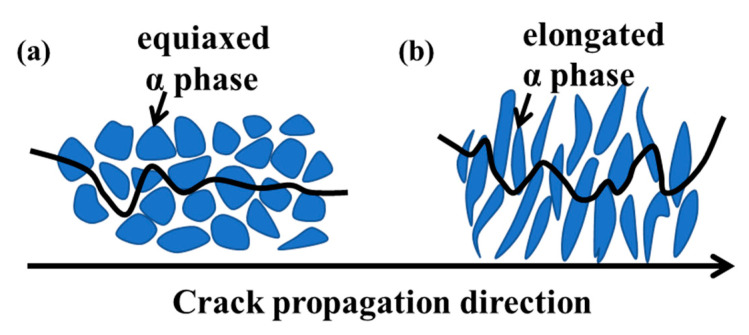
Schematic of the effect of microstructure on the crack propagation behavior for the (**a**) RD-TD specimen and (**b**) RD-ND specimen.

**Figure 11 materials-13-04332-f011:**
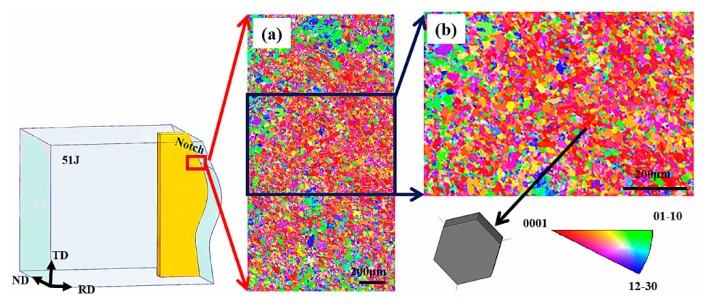
(**a**) Representative inverse pole figure (IPF) maps of the RD-TD specimen (IPF//ND); (**b**) enlarged view of the black box marked in (**a**).

**Figure 12 materials-13-04332-f012:**
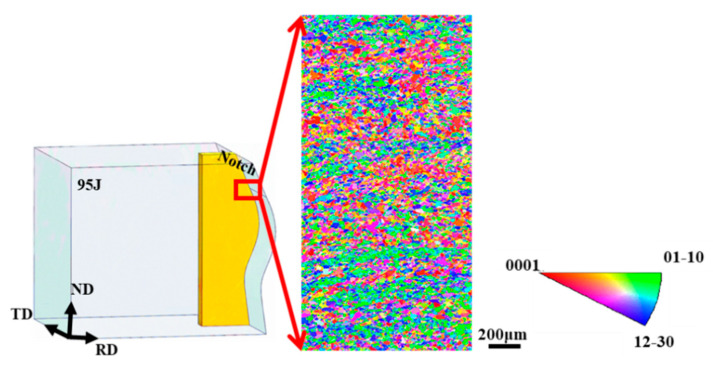
Representative IPF maps of the RD-ND specimen (IPF//TD).

**Figure 13 materials-13-04332-f013:**
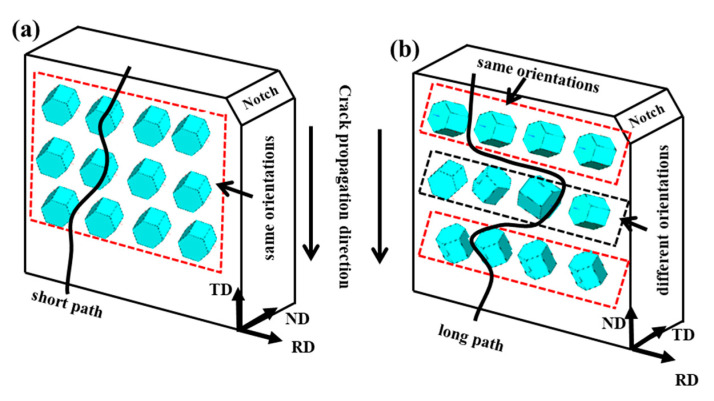
Schematic of the orientation of the α phase at different fracture specimens: (**a**) RD-TD specimen; (**b**) RD-ND specimen.

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
