# Peer review of "Impact Toughness Anisotropy of TA31 Titanium Alloy Cylindrical Shell after Ring Rolling"

_materials, 2020, doi:10.3390/ma13194332_

Round 1

Reviewer 1 Report

This is an interesting article where the authors explore impact toughness anisotropy of titanium alloy. The article is well written and of great scientific and practical interest. The main result of the work is that cracks propagate more easily in the equiaxial alpha phase than in the elongated alpha phase. The reviewer recommends the article for publication after addressing the following minor concern.

It is clear that the formation of the texture of the alloy is determined by the history of its deformation. In this sense, the purposeful formation of the structure in order to obtain the optimal mechanical characteristics of the product by selecting a specific deformation passes is considered in [Y.Estrin et al., Design of Architectured Materials Based on Mechanically Driven Structural and Compositional Patterning, Advanced Engineering Materials, Volume 21, Issue 9, 2019, 1900487; R.Kulagin et al., Benefits of pattern formation by severe plastic deformation, Applied Materials Today, Volume 15, 2019, Pages 236-241.]. Deformation path optimization is also possible in this case: upsetting, punching, saddle forging, core bar stretching and ring rolling, by introducing additional shear deformations. Also, the use of a composite initial specimen, for example, from titanium alloys Ti-6Al-4V and TA31, which, after deformation, can lead to interesting structures (textures) and, accordingly, increased impact toughness. The reviewer recommends to add the mentioned references in the Introduction.

Author Response

Great thanks for the reviewer’s meaningful comment and suggestion, which help us to improve the paper quality a lot. We have carefully revised the manuscript and provided the detailed response to each comment and suggestion. Meanwhile,the language of the manuscript has been carefully edited and the revised texts have been marked in red. We hope the revised manuscript will satisfy the reviewer and meet the journal’s requirement.

Reviewer 2 Report

Dear authors, the work, in fact, is focused on the discussion on fractographical analysis of the images you provided. In my opinion it would be worth to improve the content by the chemical\elemental analyses of the obtained structures. In general the material is interesting and by the mentioned improvement it could be published.

Author Response

(The authors gave the same response as above.)

Reviewer 3 Report

In this paper the impact toughness of TA31 titanium alloy cylindrical shell had been investigated systemically after ring rolling. The authors stated that the impact toughness of specimens with different notch orientations shows obvious anisotropy.  They found that the cracks are easier to propagate in equiaxed α phase than the elongated α phase. The α phase on the RD-TD  plane was isotropic, which maked the cracks easy to propagate along α grains in the same direction. The α phase on the RD-ND plane has a layered characteristic, and the direction of the α phase varies from layer to layer, therefore it required higher energy for crack to propagate across this layered α phase.

The presented research results of the authors are relatively interesting. The results of study are understandable and clearly described in manuscript and the number of citations is includes relatively up-to-date publications (I propose to add a few publications from the last five years).

Before publishing should be consider the following comments:

  1. The abstract contains not enough information. The authors not specified clearly the purpose of the study, methods applied and main results of study obtained.
  2. The conclusion chapter needs to be slight modification. Please add the information that effect of orientation on crack propagation behavior during impact toughness tests was performed by EBSD analysis.

Author Response

(The authors gave the same response as above.)

Round 2

Reviewer 2 Report

Dear authors,

you improved the text. The material could be published in its present form.